# Mind the Gap: Automated Corpus Creation for
# Enthymeme Detection and Reconstruction in Learner Arguments

**Maja Stahl**
Leibniz University Hannover
m.stahl@ai.uni-hannover.de

**Nick Düsterhus**
Paderborn University
nduester@mail.upb.de

**Mei-Hua Chen**
Tunghai University
mhchen@thu.edu.tw

**Henning Wachsmuth**
Leibniz University Hannover
h.wachsmuth@ai.uni-hannover.de

---

**Essay Topic:** Discuss the advantages and disadvantages of banning smoking in restaurants.

---

**Enthymematic Argument:** To smoke used to be considered a fancy act, although nowadays the society has a different point of view, smoking is wrong. [MASK] That is the reason why the government of many countries is banning smoking in private places, including restaurants. There are advantages and disadvantages of this particular prohibition.

---

**Enthymeme:** Smoking is not healthy to anyone.

---

Table 1: An enthymematic learner argument created automatically by our approach. The position of the removed ADU in the original learner argument is marked by [MASK]. The original learner argument was taken from the ICLEv3 corpus (Granger et al., 2020).

## Abstract

Writing strong arguments can be challenging for learners. It requires to select and arrange multiple argumentative discourse units (ADUs) in a logical and coherent way as well as to decide which ADUs to leave implicit, so called *enthymemes*. However, when important ADUs are missing, readers might not be able to follow the reasoning or understand the argument's main point. This paper introduces two new tasks for learner arguments: to identify gaps in arguments (*enthymeme detection*) and to fill such gaps (*enthymeme reconstruction*). Approaches to both tasks may help learners improve their argument quality. We study how corpora for these tasks can be created automatically by deleting ADUs from an argumentative text that are central to the argument and its quality, while maintaining the text's naturalness. Based on the ICLEv3 corpus of argumentative learner essays, we create 40,089 argument instances for enthymeme detection and reconstruction. Through manual studies, we provide evidence that the proposed corpus creation process leads to the desired quality reduction, and results in arguments that are similarly natural to those written by learners. Finally, first baseline approaches to enthymeme detection and reconstruction demonstrate the corpus' usefulness.

## 1 Introduction

Argumentative writing is an essential skill that can be challenging to acquire (Ozfidan and Mitchell, 2020; Dang et al., 2020; Saputra et al., 2021). To write a logically strong or cogent argument, a writer has to present sufficient evidence (i.e., the argument's premises) that justifies the argument's claim (i.e., its conclusion). In addition, arguments draw on premises that are left implicit, so called *enthymemes* (Feng and Hirst, 2011), which may be key to match the expectations of the audience (Habernal et al., 2018). However, if a relevant premise is missing, the argument may be insufficient to accept its conclusion (Johnson and Blair,

2006). Similarly, the quality of an argument will also often be limited if the conclusion is missing, as this easily renders unclear what the argument claims (Alshomary et al., 2020b).

Studies show that learners often fail to provide solid evidence for their claims (Ka-kan-dee and Kaur, 2014) as well as to compose their arguments in a coherent way (Dang et al., 2020). NLP may help learners improve the quality of their arguments by providing feedback on the arguments' logical quality. For example, the feedback could point to enthymematic gaps in arguments and make suggestions on how to fill these gaps. Until now, research on assessing logical quality dimensions of learner argumentation has focused on the amount of evidence an essay provides (Rahimi et al., 2014), and the sufficiency of its arguments, i.e., whether the premises together give enough reason to accept the conclusion (Stab and Gurevych, 2017b; Gurcke et al., 2021). To the best of our knowledge, no approach explicitly studies the adequacy of enthymemes in learner arguments yet.

To initiate research in this direction, we introduce two new tasks on learner arguments: *enthymeme detection* and *enthymeme reconstruction*. We define an enthymeme broadly here as any miss-

ing argumentative discourse unit (ADU) that would complete the logic of a written argument. The goal of the first task is to assess whether there is a possibly problematic enthymematic gap at a specified position of an argument. Given an argument with such a gap, the goal of the second task is then to generate a new ADU that fills the gap. Approaches to both tasks may help learners improve the quality of their arguments by pointing to locations of enthymemes and making suggestions on how to reconstruct the enthymemes if desired.

Data for studying enthymemes in learner arguments is neither available so far, nor is it straightforward to write respective arguments in a natural way. Therefore, we study how to automatically create a corpus for the two tasks from existing argument corpora. In particular, we propose a self-supervised process that carefully removes ADUs from high-quality arguments while maintaining the text's naturalness. We rank candidate ADUs for removal by both their contribution to an argument's quality and their centrality within an argument, combining essay scoring methods (Wachsmuth et al., 2016) with a PageRank approach (Page et al., 1999). Furthermore, we ensure the naturalness of the remaining argument (with an enthymematic gap) by utilizing BERT's capabilities to predict whether a sentence follows another sentence (Devlin et al., 2019). We create enthymemes only that cannot be detected by a BERT model. Table 1 shows an example of an enthymeme created by our approach.

Based on the third version of the ICLE corpus of argumentative learner essays (Granger et al., 2020), we automatically create 40,089 argument instances for enthymeme detection as well as 17,762 argument instances for enthymeme reconstruction. An evaluation of the created corpus suggests that our enthymeme creation approach leads to the desired quality reduction in arguments. In addition, manual annotators of the argument's naturalness deemed our modified arguments similarly natural to the original ones hand-written by the learners. Finally, we develop baseline approaches to enthymeme detection and enthymeme reconstruction based on DeBERTa (He et al., 2020) and BART (Lewis et al., 2020), respectively. Experimental results provide evidence for the usefulness of the corpus for studying the two presented tasks.

Altogether, this paper's main contributions are:[1]

---

[1]The code to reproduce the data and results can be found under: https://github.com/webis-de/EMNLP-23

- An automated approach for creating corpora of enthymematic arguments in learner essays

- A corpus for studying two new NLP tasks on learner arguments: enthymeme detection and enthymeme reconstruction

- Baseline approaches to detecting and reconstructing enthymemes computationally

## 2  Related Work

Originally, Aristotle (ca. 350 BCE / translated 2007) referred as *enthymeme* to a specific presumptive argument scheme in which certain premises are left unstated intentionally. Nowadays, the term is used both for arguments with one or more missing or implicit premises (Walton et al., 2008) and for the implicit premises themselves (Feng and Hirst, 2011). We adopt the latter use here and also cover implicit conclusions, which are often left unstated for rhetorical reasons (Al Khatib et al., 2016).

The detection of enthymemes has been studied in empirical analyses (Boltužić and Šnajder, 2016) and with computational methods (Rajendran et al., 2016; Saadat-Yazdi et al., 2023). Also, their reconstruction with generation methods has come into focus recently (Alshomary et al., 2020b; Oluwatoyin et al., 2020; Chakrabarty et al., 2021). For a specific type of enthymemes, namely *warrants* which explain the connection between premises and conclusions, Habernal et al. (2018) created a dataset with 1,970 pairs of correct and incorrect instances to be used for comprehending debate portal arguments. In contrast, we target learner arguments in this work, particularly to enable research on the question how to identify *inadequate* enthymemes, that is, those that should not be left implicit.

Argumentative writing poses many challenges for learners, particularly in the organization and development of arguments (Dang et al., 2020; Ozfidan and Mitchell, 2020). Learners commonly struggle with unclear, irrelevant, or missing thesis statements (Ozfidan and Mitchell, 2020) and face difficulties providing substantial evidence as argument premises to support their opinions (Ka-kandee and Kaur, 2014; Ozfidan and Mitchell, 2020). In some cases, learners address the claims but fail to adequately support them by omitting certain reasons (Ozfidan and Mitchell, 2020). Formulating clear conclusions, also known as the *claims* of arguments, is another area where learners often falter (Lee and Deakin, 2016; Skitalinskaya and Wachsmuth, 2023). These findings indicate that

inadequate enthymemes are a problem in learner arguments, which, so far, has not been extensively explored using computational methods.

Argument mining, the automated process of extracting argumentative discourse units (ADUs) and their relationships, has demonstrated its effectiveness in analyzing argumentative essays written by learners (Stab and Gurevych, 2014, 2017a). We do not study argument mining, but we leverage it to filter candidate enthymemes in our corpus creation approach. In line with Chen et al. (2022), we consider the ADU types *premise*, *claim*, *major claim*, and *non-argumentative*.

Moreover, we employ argument quality assessment in the corpus creation process (Wachsmuth et al., 2016). Essay argumentation has been evaluated across various quality dimensions such as an essay's organization (Persing et al., 2010), the clarity of its thesis (Persing and Ng, 2013), the strength of its arguments altogether (Persing and Ng, 2015), and the sufficiency of each individual argument (Stab and Gurevych, 2017b). This has also facilitated the development of approaches to generate missing argumentative ADUs, specifically argument conclusions (Gurcke et al., 2021). In a related line of research, approaches have been proposed to suggest specific revisions of argumentative essays (Afrin and Litman, 2018), to assess the quality of revisions (Liu et al., 2023), and to perform argument revisions computationally (Skitalinskaya et al., 2023). Our proposed enthymeme-related tasks complement these attempts, having the same ultimate goal, namely to learn support systems for argumentative writing (Wambsganss et al., 2020).

In conclusion, the computational study of enthymemes in learner arguments is a fairly unexplored research area, and the automatic detection and reconstruction of inadequate enthymemes in this domain have not been addressed as of now. Developing corpora specifically designed to analyze inadequate enthymemes in learner arguments may provide valuable insights into common pitfalls and enable the development of effective methods for helping learners to improve their argumentative writing skills in this regard.

## 3 Automatic Corpus Creation

This section presents our approach to automatic corpus creation. Given a base corpus of argumentative learner texts as input, we propose to create enthymematic arguments automatically by remov-

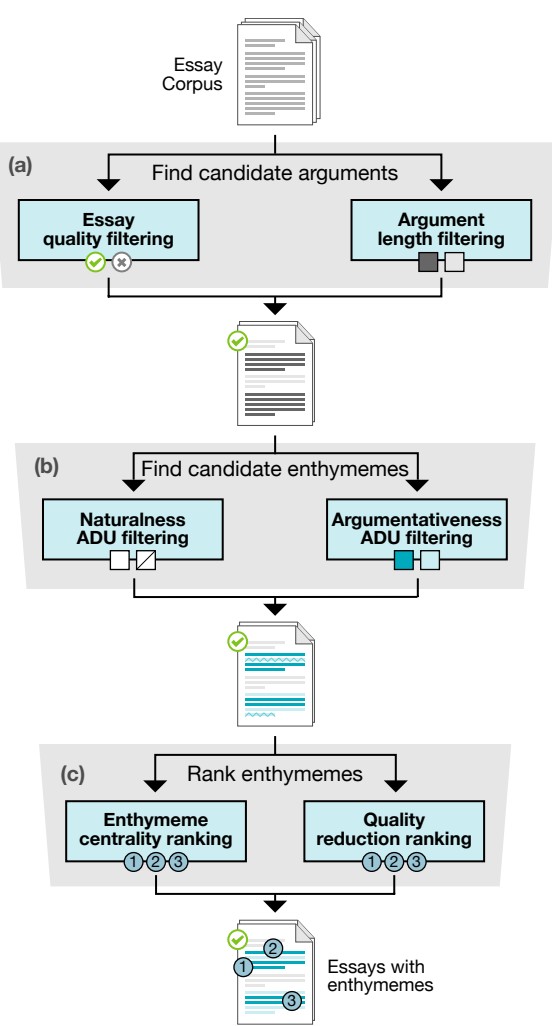

Figure 1: Proposed corpus creation process: (a) Arguments of a defined minimum length from high-quality essays are candidates for enthymeme creation. (b) Argumentative ADUs whose removal maintains naturalness are candidates for enthymemes. (c) The candidate ADUs are ranked by centrality and quality reduction to determine the final enthymeme.

ing argumentative discourse units (ADUs) from high-quality arguments in these texts. In the following, we detail the three parts that constitute our approach: (a) finding candidate arguments, (b) finding candidate enthymemes in these arguments, and (c) ranking the candidates to remove the most suitable one per argument. Figure 1 illustrates the corpus creation process.

### 3.1 Finding Candidate Arguments

We consider only arguments (i) from essays of high quality that (ii) have a specified minimum length. With these filtering steps, we aim to minimize the risk of already existing, unknown enthymemes within an argument.

| | Organization | | Clarity | | Strength | |
|---|---|---|---|---|---|---|
| **Approach** | mae | mse | mae | mse | mae | mse |
| Our approach | .320 | .165 | .498 | .395 | **.370** | **.217** |
| Persing et al. | .323 | .175 | **.483** | **.369** | .392 | .244 |
| Wachsmuth et al. | **.314** | **.164** | .501 | .425 | .378 | .226 |

Table 2: Performance of our essay scoring models for *organization*, thesis *clarity* and argument *strength* in terms of mean absolute error (mae) and mean squared error (mse). Lower values are better. Our models are close to or better than the state-of-the-art approaches.

**Essay Quality Filtering**    We model three essay-level quality dimensions that have been well studied in previous work: *organization*, *thesis clarity*, and *argument strength* (Persing et al., 2010; Persing and Ng, 2013, 2015; Wachsmuth et al., 2016; Chen et al., 2022). To assess these dimensions for the base essays, we train one LightGBM regression model (Ke et al., 2017) per quality dimension, using a combination of new features and reimplemented features from Wachsmuth et al. (2016):[2]

- ADU $n$-grams, with $n \in \{1, 2, 3\}$
- Sentiment flow (Wachsmuth et al., 2015)
- Discourse function flow (Persing et al., 2010)
- Prompt similarity flow
- Token and POS $n$-grams
- Length statistics
- Frequency of linguistic errors (*LanguageTool*)
- Distribution of named entity types
- Distribution of metadiscourse type markers (Hyland, 2018)

We performed ablation tests to determine the best feature combination per quality dimension in terms of mean squared error (mse). Details on all features and the best combinations can be found in Appendix A. Table 2 shows the performance of our essay scoring models with the best-performing feature combinations. Our models are on par with the above-mentioned state of the art approaches, even being best in predicting argument strength.

For filtering, we average the three scores per essay to an overall score between 1.0 (very bad) to 4.0 (very good) (Persing and Ng, 2015) and keep

only the essays with an overall score above 3.0, as they should be of decent quality.

**Argument Length Filtering**    As a simple heuristic, we expect essay paragraphs with at least four sentences to be arguments, since we observed that shorter paragraphs are unlikely to be substantial enough to contain a full argument. Furthermore, removing one ADU from an argument with three or fewer sentences may likely corrupt the argument beyond creating an inadequate enthymeme.[3]

### 3.2 Finding Candidate Enthymemes

From the candidate arguments, we select ADUs as candidate enthymemes in two ways: (i) We ensure that removing the ADU does not destroy the argument's naturalness, that is, the arguments still read like they were written by a human; and (ii) we require an ADU to be argumentative in order to be selected as a candidate enthymeme. Otherwise, it might not reduce the logical quality of an argument.

**Naturalness ADU Filtering**    We aim to create enthymemes that are likely to appear in human-written arguments. To prevent the remaining argument from being unnatural, we check whether a candidate enthymeme can be detected by BERT's next sentence prediction (Devlin et al., 2019): We retain only those candidates for which a BERT model predicts that the sentences before and after it could be neighbors, that is, the enthymeme does not disrupt the text's naturalness. The text naturalness is evaluated manually in Section 5.2.

**Argumentativeness ADU Filtering**    Removing a non-argumentative ADU from an argument does not necessarily create an inadequate enthymeme. Thus, we predict the ADU type of argument sentences adopting the approach of Chen et al. (2022), which distinguishes between *premise*, *claim*, *major claim*, and *non-argumentative*. Additionally, we consider only ADUs with more than five tokens significant enough to an argument to create an inadequate enthymeme based on manual inspection.

### 3.3 Ranking Candidate Enthymemes

To find the most suitable candidate enthymemes, we use a PageRank-based approach (Page et al., 1999) that ranks ADUs according to two criteria:

---

[2]Initial experiments on predicting the essay quality dimensions using Longformer (Beltagy et al., 2020) led to lower performance compared to the feature-based approaches, prompting us to stop further investigations in this direction.

[3]We also only create enthymematic arguments of at most 500 tokens so that together with added special tokens, they fit the maximum input length of most common language models.

(i) their centrality within an essay and (ii) their contribution to the essay's quality. We hypothesize that this enables us to choose high-quality ADUs for removal that are critical to an argument and therefore leave a quality-reducing, inadequate enthymeme. We take the whole essay into consideration to have as much context as possible for estimating the centrality and quality contribution of the ADUs. This way, we can also reuse our essay scoring models. After ranking, we determine the highest-ranked candidate enthymeme on argument level.

**Enthymeme Centrality Ranking**  Previous work has modeled a document as a graph where each node corresponds to a sentence, and the weights are derived from sentence similarity. This allows estimating the centrality of the sentences by calculating the sentence rank (Erkan and Radev, 2004; Mihalcea and Tarau, 2004; Alshomary et al., 2020a). We adapt this idea by constructing a graph that represents an essay's sentences and title by encoding each sentence $s_i$ as a vector $\mathbf{s}_i$ using sBERT embeddings (Reimers and Gurevych, 2019). From this, we construct a similarity matrix $A'$ with

$$a'_{ij} := \cos(\mathbf{s}_i, \mathbf{s}_j) = \frac{\mathbf{s}_i \cdot \mathbf{s}_j}{||\mathbf{s}_i|| \cdot ||\mathbf{s}_j||}. \quad (1)$$

Since cosine similarity lies in the interval $[-1, 1]$, we apply a softmax on each row $A'_i$ to obtain a stochastic matrix $A$, with $A_i := softmax(A'_i)$.

**Quality Reduction Ranking**  We aim to remove sentences that contribute to the quality of an argument in the sense that the created enthymemes harm the quality. Since there is, to our knowledge, no research on the quality contributions of individual sentences in the context of an essay, we estimate it by removing the sentences and observing the change in essay scores. In more detail, given the essay score $q_{\{1,...,n\}}$ of an essay with $n$ sentences, we estimate the quality contribution $c_i$ of the $i-$th sentence to be:

$$c_i := q_{\{1,...,n\}} - q_{\{1,...,i-1,i+1,...,n\}} \quad (2)$$

We use our essay scoring models to determine the quality scores for each candidate enthymeme. For the other sentences, we use the average quality score of all sentences in the essay.

As for the sentence centrality, we construct a quality graph, represented as matrix $B'$, where each sentence $s_i$ is linked to each other sentence $s_j$ with the weight being the quality contribution of $s_j$, i.e.

$b'_{ij} := c_j$. To align the value range of the quality scores with the centrality scores, we use min-max scaling before applying the softmax function:

$$B_i := softmax(\frac{B'_i - \min(B'_i)}{\max(B'_i) - \min(B'_i)}) \quad (3)$$

**PageRank-based Ranking**  To rank the sentences by both criteria, their centrality and their quality contribution, we combine the matrixes $A$ and $B$ into one transition matrix $M$:

$$M := 0.5 \cdot A + 0.5 \cdot B \quad (4)$$

The final ranking is obtained by applying the power iteration method. Accordingly, the sentences with the highest rank are expected to be those that balance high centrality within the essay and great contribution to the quality of the essay.

## 4  Corpus

This section summarizes the corpora that we use for model training and for automatic corpus creation. Then, we give details about the composition and statistics of the resulting created corpus.

### 4.1  Base Corpora

To train essay scoring models, we needed essay quality data. For this, we reused the 1,003 argumentative learner essays rated for organization by (Persing et al., 2010), 830 essays rated for thesis clarity (Persing and Ng, 2013), and 1,000 essays rated for argument strength (Persing and Ng, 2015). These essays come from the second version of the International Corpus of Learner English, ICLEv2 (Granger et al., 2009).

The basis for our automatically-created corpus is the third version of the same corpus, ICLEv3 (Granger et al., 2020). It contains 8,965 argumentative essays written by learners with 25 different language backgrounds.

### 4.2  Corpus Composition and Statistics

Each filtering step described above resulted in the number of essays, arguments, and candidate enthymemes shown in Table 3. The steps applied filters to essays, arguments, or ADUs directly, which mostly resulted in indirect effects on other levels.

With a probability of 80%, we removed the highest-ranked ADU of an argument with at least one candidate enthymeme to create two instances: (a) One positive example with a separator token marking the correct enthymeme position and (b)

| Data | # Essays | # Arguments | # Candidates |
|---|---|---|---|
| Base Corpus, ICLEv3 | 8,965 | 63,811 | 289,779 |
| After filtering by | | | |
| – essay quality | 5,541 | 36,835 | 176,580 |
| – argument length | 5,464 | 22,526 | 146,567 |
| – naturalness | 5,464 | 22,525 | 137,855 |
| – argumentativeness | 5,464 | 22,331 | 110,702 |

Table 3: Results of the four filtering steps of our corpus creation approach to the ICLEv3 corpus: Number of essays, arguments and candidate ADUs for enthymeme creation after each filtering step.

one negative example with the separator token at a random, incorrect position. In the remaining cases, the original argument without deletion and with a separator token at a random position is added as a negative example to improve model robustness. This results in 17,762 positive and 22,327 negative examples, with 40,089 in total. Within the positive examples, 63.28% of the enthymemes are premise removals, 30.97% claim removals, and 5.75% major claim removals, according to the ADU type classifiers based on Chen et al. (2022)'s work. We randomly split the corpus into training, validation, and test set using a ratio of 7:1:2.

To provide insights into the behavior of our approach, Figure 2 visualizes the enthymeme positions in our corpus compared to a randomly-generated corpus created by removing random argumentative ADUs with more than five tokens (*argumentativeness ADU filtering* only). The distribution of enthymeme positions shows a slight preference for the first ADU, which often introduces the topic and stance and is central to the argument. The positional bias can largely be attributed to differences in argument length.

## 5 Corpus Analysis

This section reports on our evaluation of the effectiveness of our automatic approach for creating inadequate enthymemes that are harmful to the quality of arguments while not making their texts seem unnatural. To this end, we manually assessed (a) the induced reduction in logical argument quality in terms of sufficiency and (b) the naturalness of the enthymematic arguments in comparison to the original arguments written by learners.

### 5.1 Manual Sufficiency Evaluation

In argumentation theory, sufficiency captures whether an argument's premises together make it

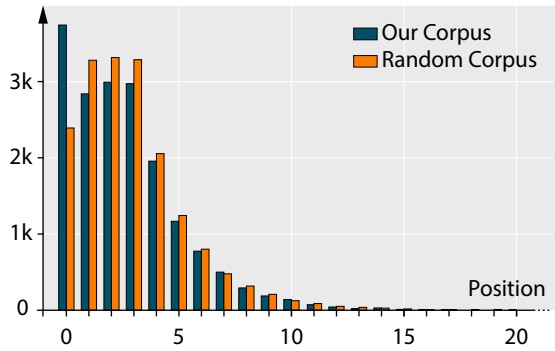

Figure 2: Histogram of the chosen enthymeme positions in our corpus and in a baseline corpus with enthymemes at random positions. Our approach more often chooses the first ADU to create the enthymeme, but the rest of the distribution is comparable to the random corpus.

rationally worthy of drawing its conclusion (Johnson and Blair, 2006), which captures our intended goal of having only adequate enthymemes well. We relax the notion of sufficiency here to also account for missing argument conclusions, and we deem an argument sufficient if (1) its premises together give enough reason to accept the conclusion *and* (2) its conclusion is clear. Given this definition, we evaluate the change in sufficiency resulting from the created enthymemes, that is, whether it is *more*, *equally*, or *less* sufficient than the original argument. Recall that the goal of our approach is to create *insufficient* arguments.

For evaluation, we randomly chose 200 original arguments from the given corpus and their modified versions created with four different methods:

(a) Our full approach

(b) Our approach without centrality ranking (i.e., $M := B$ instead of Equation 4)

(c) Our approach without quality reduction ranking (i.e., $M := A$)

(d) A baseline that randomly removes any argumentative ADU with at least five tokens[4]

We hired five native or bilingual English speakers via *Upwork* for this task and paid 13$ per hour of work. The annotation guidelines can be found in Appendix B. To model each annotator's reliability, we use MACE for combining the non-expert annotators' labels into final labels (Hovy et al., 2013).

Table 4 shows the distribution of MACE labels. The results suggest that all compared approaches

---

[4]The baseline equals the modified arguments from the randomly created corpus, (*Random Corpus*).

| | Sufficiency | | |
|---|---|---|---|
| **Arguments modified by** | more ↓ | equal | less ↑ |
| Full approach | **10.0%** | 29.0% | **61.0%** |
| – w/o centrality ranking | 13.5% | 39.0% | 47.5% |
| – w/o quality reduction ranking | 10.5% | 36.0% | 53.5% |
| Random baseline | 10.5% | 36.0% | 53.5% |

Table 4: Manual sufficiency evaluation: Percentage of arguments that are more/equally/less sufficient after creating enthymemes in arguments with our full approach, two variations of our approach, and random argumentative ADU removal. The best value per column is marked in bold. Our approach succeeds in reducing sufficiency.

reduce the arguments' sufficiency in most cases, which is expected from the definition of sufficiency. However, our full approach works best in decreasing the original argument's sufficiency, outperforming its two variations and the random baseline. Additionally, our full approach to enthymeme creation is least likely to cause an increase in sufficiency. The average pairwise inter-annotator agreement in terms of Kendall's $\tau$ is 0.25.

### 5.2 Manual Naturalness Evaluation

To assess the impact of our naturalness ADU filtering (Section 3.2), we manually evaluated the naturalness of the arguments with enthymematic gaps in comparison to the original learner arguments. For this, three authors of this paper manually inspect a random sample of 100 arguments: 50 original learner arguments and 50 arguments modified by our full approach. We scored naturalness on a 3-point scale:

1. *Artificial.* A part of the argument is unrelated to the preceding or succeeding text. For example, a reference cannot be resolved (cohesion), the text abruptly changes its theme or does not have any continuous theme (coherence), or some definitely required information is missing (clarity).

2. *Partly Natural.* The text is mainly coherent, its use of cohesive devices does not affect continuity or comprehensibility, and the information provided suffices to get the author's point.

3. *Natural.* The text is coherent, its use of cohesive devices is correct, and the information provided allows to clearly understand the author's point.

| | Naturalness Scores | | | |
|---|---|---|---|---|
| **Arguments** | 1 | 2 | 3 | Mean ↑ |
| Original | 18%  (9) | 36% (18) | 46% (23) | 2.28 |
| Modified | 20% (10) | 38% (19) | 42% (21) | 2.22 |

Table 5: Manual naturalness evaluation: Distribution of majority scores and their mean for the *original* arguments written by learners and arguments *modified* by our approach. Naturalness is largely maintained.

The results of the manual naturalness evaluation are shown in Table 5. Both original and modified arguments achieve a mean score slightly above partial naturalness. This indicates that the arguments written by learners already have some naturalness issues. Additionally, creating enthymematic gaps using our approach only slightly reduces the arguments' naturalness. The average pairwise inter-annotator agreement in terms of Kendall's $\tau$ is 0.27.

## 6 Experiments

Our automatically-created corpus is meant to enable studying two new tasks on learner arguments: Enthymeme detection and enthymeme reconstruction. This section presents baselines for the tasks in order to demonstrate the usefulness of the corpus.

### 6.1 Enthymeme Detection

We treat enthymeme detection as a binary classification task: Given a learner argument and a position within the argument, predict whether there is an inadequate enthymeme (say, a logical gap) at the given position. We mark the positions in question using a separator token, [SEP]. Furthermore, we use the token type IDs to differentiate between the sequences before and after the separator token.

**Models** For classification, we fine-tune the language model DeBERTa (He et al., 2020) with 134M parameters, 12 layers, and a hidden size of 768 (*microsoft/deberta-base*) provided by Huggingface (Wolf et al., 2020). To quantify its learning success, we also report the results of a random classifier that makes random predictions and a majority classifier that always predicts the majority training class.

**Experimental Setup** We tune the training hyperparameters of DeBERTa in a grid-search manner using Raytune (Liaw et al., 2018). We explored numbers of epochs in $\{8, 16, 24\}$ with learning rates in $\{10^{-6}, 5 \cdot 10^{-6}, 10^{-5}\}$ and batch size 16. The

| Approach | Accuracy | Precision | Recall | $F_1$-Score |
|---|---|---|---|---|
| Random | 0.51 | 0.48 | 0.51 | 0.50 |
| Majority | 0.53 | 0.00 | 0.00 | 0.00 |
| DeBERTa | **0.72** | **0.73** | **0.64** | **0.68** |

Table 6: Enthymeme detection results: Classification performance of a fine-tuned DeBERTa classifier compared to a random baseline and a majority baseline.

best $F_1$-score on the validation set was achieved by training for 24 epochs with a learning rate of $10^{-5}$.

**Results**   Table 6 shows the classification results. The learning success of the DeBERTa model suggests the possibility of automating the task of enthymeme detection in learner arguments on our corpus. However, further improvements using more advanced approaches are expected and encouraged.

### 6.2 Enthymeme Reconstruction

We treat enthymeme reconstruction as a mask-infilling task: Given a learner argument and a known enthymeme position within the argument, generate the enthymeme. We mark the enthymeme position with a special token, [MASK]. For this task, corpus instances without enthymemes are excluded since they lack target text.

**Models**   For mask-infilling, we fine-tune BART models (Lewis et al., 2020) with 400M parameters, 24 layers, and a hidden size of 1024 (*facebook/bart-large*) provided by Huggingface (Wolf et al., 2020). We compare two different approaches: First, we adapt the approach of Gurcke et al. (2021) to argument conclusion generation to our task by providing the model with the argument and the mask token at the enthymeme position as input. To evaluate the importance of contextual information, the second approach, *BART-augmented*, extends the approach of Gurcke et al. (2021) by prepending the essay topic and the predicted ADU type of the enthymeme to the input.[5]

**Experimental Setup**   Using Huggingface's (Wolf et al., 2020) default sequence-to-sequence training parameters and learning rate $2 \cdot 10^{-5}$, we fine-tune the BART models for five epochs. For this task we did not perform extensive hyperparameter optimization since the models' performance on the validation set was already comparable to related work (Gurcke et al., 2021) and sample experiments

---

[5]ADU types are predicted with the classifiers that we trained based on the work by Chen et al. (2022).

| Approach | BERTSc. | ROU.-1 | ROU.-2 | ROU.-L |
|---|---|---|---|---|
| Gurcke et al. (2021) | 0.247 | 21.50 | 4.62 | 17.10 |
| BART-augmented | **0.252** | **21.83** | **4.64** | **17.28** |

Table 7: Enthymeme reconstruction results: Rescaled BERTScore $F_1$ and ROUGE-1/-2/-L for Gurcke et al. (2021) and the extended approach, BART-augmented.

| # | Data | Source |
|---|---|---|
|  | **[no enthymeme position]** | DeBERTa |
| 1 | One argument that the students use the credit card unwisely. | Human learner |
|  | **[enthymeme position]** | DeBERTa |
| 2 | *According to <R> the students need to pay off the debts in short period of time.* | BART-augmented |
| 3 | They need to do part-time-jobs to earn money to settle the debts. | Human learner |
|  | **[no enthymeme position]** | DeBERTa |
| 4 | The students invest much of time to do part-time jobs therefore waste a lot of time on their studies. | Human learner |
|  | **[no enthymeme position]** | DeBERTa |
| 5 | The part-time jobs which is lower payment to the students. | Human learner |
|  | **[no enthymeme position]** | DeBERTa |
| 6 | The student should manage time effectively to develop a balanced and health life style. | Human learner |
|  | **[enthymeme position]** | DeBERTa |
| 7 | *In addition, the students need to learn how to manage their financial affairs in a responsible way.* | BART-augmented |

Table 8: Illustration of the output from our detection model (*DeBERTa*) and our reconstruction model (*BART-augmented*): Application to each potential position in an example learner argument from the ICLEv3 (Granger et al., 2020) about the use of credit cards by students. The result is an argument with seven sentences, of which two are reconstructed enthymemes (#2 and #7).

with other hyperparameter values did not lead to further improvements.

**Results**   Table 7 reports the automatic evaluation results for enthymeme reconstruction. We only include test instances for which both models generated text. In line with Gurcke et al. (2021), we report the recall-oriented ROUGE scores (Lin, 2004), since we see it is worse to omit parts of the target text than to generate text that goes beyond the ground-truth text for this task. The results suggest that the additional information on the topic and/or target ADU type can help with enthymeme

reconstruction, although the gain is small on average. Moreover, a manual investigation of 30 arguments, as detailed in Appendix D, suggested that language models can learn to automatically reconstruct enthymemes in learner arguments using our corpus. We expect more advanced approaches can and should further increase effectiveness.

To provide further insights into how our models work on full arguments we applied our enthymeme detection model to each potential position in an example learner argument from the ICLEv3 (Granger et al., 2020). On positions that were classified as enthymematic, we applied our enthymeme reconstruction model to generate the missing component. The results are displayed in Table 8. Our models identified two enthymeme positions and generated corresponding texts, that make the connection between credit card usage and depts and the implicit claim of the argument towards the topic explicit.

## 7 Conclusion

Until now, data for studying enthymemes in learner arguments is neither available nor is it straightforward to write respective arguments in a natural way. Therefore, this paper has studied how to automatically create corpora of enthymematic learner arguments using a self-supervised process that carefully removes ADUs from argumentative texts of high-quality while maintaining the texts' naturalness. To be able to point to possibly problematic gaps in learner arguments and make suggestions on how to fill these gaps, we present the tasks of enthymeme detection and reconstruction for learner arguments that can be studied using the created corpus.

Our analyses provided evidence that our corpus creation approach leads to the desired reduction in logical argument quality while resulting in arguments that are similarly natural to those written by learners. Hence, we conclude that, mostly, inadequate enthymemes are created in the sense of missing ADUs that would complete the logic of a written argument. Moreover, first baselines demonstrate that large language models can learn to detect and fill these enthymematic gaps when trained on our corpus. Such models may help learners improve the logical quality of their written arguments.

## 8 Limitations

Aside from the still-improvable performance of the presented baseline models and the limitations of the models that are used as part of our approach, we see three notable limitations of the research presented in this paper. We discuss each of them in the following.

First, due to the lack of learner argument corpora with annotated inadequate enthymemes, we cannot compare the automatically-created enthymemes to human-created ones. Creating such enthymemes intentionally in arguments is not a straightforward task for humans, which is why we aimed to do this automatically in the first place. However, it should be noted that we cannot guarantee that our created enthymemes consistently align with the traditional understanding of enthymemes as studied in argumentation theory.

Second, despite applying quality filters to the original learner essays, there is a possibility that the arguments contain unknown inadequate enthymemes. This can harm the effectiveness of the corpus annotations for both presented tasks.

Lastly, we evaluate our corpus creation approach only on a single corpus of argumentative learner essays. This particular corpus may not represent the whole diversity of argumentative texts or learner proficiency levels. Consequently, the generalizability of our findings and the applicability of the proposed approaches to a wider array of argumentative writing remains to be explored.

## 9 Ethical Considerations

We do not see any apparent risks of the approach we developed being misused for ethically doubtful purposes. However, it is important to acknowledge that automatically-created artificial data could be mistaken for real or authentic data, leading to false understandings, biased conclusions, or misguided decision-making. This is not our intended use case for the created corpus, which is meant for method development. If texts from the corpus are directly displayed to learners, we suggest to mark that they have been created automatically.

## Acknowledgments

This project has been partially funded by the German Research Foundation (DFG) within the project ArgSchool, project number 453073654. We would like to thank the participants of our study and the anonymous reviewers for the feedback and their time.

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

## A  Essay Scoring Models

This section gives additional details on the features and the model performance using different combinations of features used as input for the essay scoring models predicting *organization*, *thesis clarity*, and *argument strength*. We also report the feature combinations that led to the lowest mean squared error (mse) for each quality dimension.

### A.1  Details on Features

- *ADU n-grams*: ADU n-grams, with $n \in \{1, 2, 3\}$; frequencies of ADU type combinations and sequences of ADU types within paragraphs.

- *Sentiment flow*: Sequence of the paragraph's sentiments, using Barbieri et al. (2020)'s sentiment classifier.

- *Discourse function flow*: Sequences of discourse functions: introduction, body, and conclusion (Persing et al., 2010).

- *Prompt similarity flow*: Max., min. and mean cosine similarity of the paragraph embeddings to the prompt; Similarity flow, i.e., sequence of the paragraph's similarities to prompt using sBERT embeddings (Reimers and Gurevych, 2019).

- *Token and POS n-grams*: Token $n$-grams ($n \in \{1, 2, 3\}$) that occur in more tham 1% of the training data; POS $m$-grams ($m \in \{1, .., 5\}$) that occur in more than 5% of the training data.

- *Length statistics*: Number of paragraphs, sentences, tokens; maximum, minimum and average number of sentences per paragraph.

- *Frequency of linguistic errors*: Number of linguistic errors given by *LanguageTool*.

- *Distribution of named entity types*: Occurrences per named entity type using spaCy (Honnibal et al., 2020).

- *Distribution of metadiscourse type markers*: Occurrences per metadiscourse marker category, as proposed by Hyland (2018).

### A.2 Comparison of Feature Combinations

Table 9 compares the model performances for different combinations of input features: all features, all feature combinations in which one feature is left out and the feature combination that performed best in terms of mse for each of the three quality dimensions.

The combinations of features that led to the best model performance it terms of mse, as reported in Table 9, are:

- *Organization.* Linguistic errors, token and pos n-grams, length statistics, named entity types, prompt similarity flow.

- *Thesis Clarity.* Linguistic errors, sentiment flow, discourse function flow, named entity types, prompt similarity flow.

- *Argument Strength.* ADU n-grams, token and pos n-grams, prompt similarity flow.

## B Annotation Guidelines: Sufficiency

### B.1 Task Description

In this task, you will be presented with arguments from argumentative learner essays (original arguments) and different variations of them (modified arguments). We want to evaluate the argument quality of the variations in relation to the original argument. The task consists of reading 200 arguments and up to four argument variations each. You are asked to assess for each modified argument whether the quality of the modified argument is higher, equal, or lower than the quality of the original argument. We measure argument quality in terms of sufficiency as defined below. The estimated time for this task is 3-5 minutes to evaluate all variations of one original argument, approximately 15 hours of work.

### B.2 Data

You are asked to assess the sufficiency of the modified arguments in relation to the corresponding original argument. The modified arguments are variations of the original argument in which one of the original sentences is removed. We replaced the removed sentence with "<mask>" to indicate the deletion position. When a row has no original argument, the previous original argument is the corresponding original argument. The same holds for the topic.

### B.3 Sufficiency Definition

An argument is sufficient when (1) the premises together give enough reason to accept the conclusion, (2) and it is clear what the argument's conclusion is.

### B.4 Annotation Scale

- Sufficiency increased: +1
- Sufficiency unchanged: 0
- Sufficiency decreased: +1

## C Annotation Guidelines: Naturalness

### C.1 Task Description

For this task, you will be asked to evaluate what we defined as "naturalness" of a paragraph. Therefore you will rate the provided paragraphs on an ordinal scale from 1-3. A rating of 3 points represents the highest grade and 1 the lowest. To come to a conclusion, we would like you to pay attention to the three main criteria of coherence, cohesion,

| Features | Organization | | Thesis Clarity | | Argument Strength | |
|---|---|---|---|---|---|---|
| | mae | mse | mae | mse | mae | mse |
| All features | .325 | .171 | .521 | .419 | .375 | .222 |
| – w\o ADU n-grams | .324 | .171 | .515 | .413 | .374 | .224 |
| – w\o sentiment flow | .324 | .172 | .519 | .413 | .368 | .220 |
| – w\o discourse function flow | **.318** | .166 | .520 | .412 | .378 | .227 |
| – w\o prompt similarity flow | .324 | .174 | .551 | .466 | .379 | .227 |
| – w\o token and pos n-grams | .322 | .172 | .510 | .412 | .398 | .246 |
| – w\o named entity types | .323 | .172 | .526 | .424 | .378 | .227 |
| – w\o length statistics | .336 | .185 | .519 | .413 | .376 | .222 |
| – w\o linguistic errors | .325 | .173 | .516 | .417 | .377 | .223 |
| – w\o metadiscourse type markers | .325 | .173 | .517 | .416 | .377 | .224 |
| Feature combination with lowest mse per dimension | .320 | **.165** | **.498** | **.395** | **.370** | **.217** |

Table 9: Features for essay scoring: Model performances for different combinations of input features: All features, all features but one, and the feature combination that performed best in terms of mse for each of the three quality dimensions *organization*, *thesis clarity* and *argument strength*. The best values per column are marked in bold.

and clarity. The paragraph should be awarded 3 points if the topic is not abruptly changed, cohesive devices can be resolved and the paragraph does not lack the information we do not expect elsewhere. We award 2 points if we see stronger flaws in cohesion or the use of cohesive devices or clarity, which are not out of the ordinary for student essays. If we observe a strong discontinuity between parts of the paragraph, we award 1 point. We do not expect the essay written by English learners to be perfectly coherent, cohesive and clear. However, if you notice strong incoherences or cohesive ties that cannot be resolved, the paragraph would be seen as artificial. Intuitively, read the paragraph sentence per sentence, and examine the continuity/transition between the sequence of sentences. If you consider a sentence not dependent on the preceding text, the text can be seen as unnatural and artificial. The evaluation scheme and scoring criteria are presented below. Concrete representations of natural or artificial are difficult to precisely define, the criteria should help the annotators to judge the "naturalness" based on their feeling for the English language and knowledge about the level of student's writing.

### C.2 Annotation Scale

1. *Artificial*: A part of the paragraph (e.g. a sentence) seems unrelated to the preceding or succeeding text. For example, a reference cannot be resolved. (cohesion). Or the text abruptly changes the overall thematic scheme or does not have an underlying theme. (coherence). Or a substantial amount of information that we would have expected based on the given text cannot be found (clarity)

2. *Partially Natural*: The text is largely coherent (no strong incoherences), the use of cohesive devices does not damage the continuity/comprehensibility of the text, and the provided information is enough to get his point across.

3. *Natural*: The text is at least not incoherent, cohesive devices are correctly used, and the provided information allows one to understand his argument clearly.

## D    Manual Investigation: Enthymeme Reconstruction

As a manual investigation of the generated reconstructed enthymemes, we asked three annotators to evaluate 30 arguments along the three dimensions *clarity*, *argument strength* and *coherence* on 5-point scales as defined below, with 5 being the best score. The annotators were given arguments with a [MASK] token at the enthymeme position as context and three different ADUs for the masked position: (i) the original ADU, (ii) the ADU generated by Gurcke et al. (2021)'s approach, and (iii) the ADU generated by the approach *BART-augmented*.

### D.1    Definitions and Scales

**Clarity**    The argument has a high clarity if "it uses correct and widely unambiguous language as well as if it avoids unnecessary complexity and deviation from the issue" (Wachsmuth et al., 2017).

| Approach | Clarity | | Argument Strength | | Coherence | |
|---|---|---|---|---|---|---|
| | Mean | Agreement | Mean | Agreement | Mean | Agreement |
| Original | **3.79** | 57% | **3.64** | 70% | **3.84** | 90% |
| Gurcke et al. (2021) | 3.17 | 73% | 2.86 | 60% | 2.99 | 60% |
| BART-augmented | 3.39 | 40% | 3.13 | 60% | 3.47 | 63% |

Table 10: Manual enthymeme reconstruction results: Mean evaluation scores and majority agreement ratios for the original ADUs and ADUs generated by Gurcke et al. (2021)'s approach and the extended approach, *BART-augmented*. Best values are marked bold.

1. The addition of the ADU makes the argument less clear. The ADU itself uses incorrect language and does not align with the topic of the argument.

2. The addition of the ADU makes the argument less clear. The ADU itself is not fully understandable and only fits partly to the topic of the argument.

3. The addition of the ADU does not improve the clarity of the argument. The ADU itself is in terms of language in line with the rest of the argument.

4. The addition of the ADU slightly improves the clarity of the argument. The ADU itself uses mostly correct, unambiguous language and fits the topic.

5. The addition of the ADU improves the clarity of the argument. The ADU itself uses correct, unambiguous language and fits the topic.

**Argument Strength**    Following (Persing and Ng, 2015), we define the strength of an argument by its effectiveness in convincing a majority of a (reasonable) audience.

1. The addition of the ADU does not fit the argument, it contradicts or makes the main standpoint unclear.

2. The addition of the ADU weakens the effectiveness of the argument.

3. The addition of the ADU leads to neither an improvement in strength nor does it make the argument less convincing.

4. The addition of the ADU has a positive effect on the effectiveness of the argument.

5. The addition of the ADU strengthens the argument, it becomes more convincing.

**Coherence**    In line with (Smalley et al., 2001), we define that a coherent paragraph or argument should contain sentences that are arranged logically and have a smooth flow.

1. The addition of the ADU results in an illogical argument, which is hard to follow. The topic of the ADU does not align with the rest of the argument.

2. The addition of the ADU results in an argument that is less logical than before and the argument is more difficult to follow. The topic of the ADU does not fully fit into the rest of the argument.

3. The addition of the ADU is in line with the flow and logical arrangement of the argument. The ADU fits the underlying topic.

4. The addition of the ADU improves the coherence of the argument a little. The ADU adds to the logical arrangement and makes it easier to follow.

5. The addition of the ADU improves the coherence of the argument a lot. The ADU adds to the logical arrangement and makes it much easier to follow.

### D.2 Results

Table 10 shows the results of the manual enthymeme reconstruction evaluation. As expected, the original ADUs lead to the best average score in all three dimensions, followed by the extended approach *BERT-augmented*. The generated ADUs reach medium quality in terms of clarity, argument strength and coherence in most cases. Overall, the quality of the generated ADUs seems to be very comparable with the original ADUs written by learners, which suggests that language models can learn to automatically reconstruct enthymemes in learner arguments using our corpus.