# OpenReview forum: "Mind the Gap: Automated Corpus Creation for Enthymeme Detection and Reconstruction in Learner Arguments"
_EMNLP/2023/Conference — EMNLP 2023 Findings_

### Official Review · Reviewer_P3fJ · 2023-07-20

**Soundness:** 3

**Excitement:**

3: Ambivalent: It has merits (e.g., it reports state-of-the-art results, the idea is nice), but there are key weaknesses (e.g., it describes incremental work), and it can significantly benefit from another round of revision. However, I won't object to accepting it if my co-reviewers champion it.

**Paper Topic And Main Contributions:**

In the paper titled "Mind the Gap: Automated Corpus Creation for Enthymeme Detection and Reconstruction in Learner Arguments", the authors propose a methodology based on deep learning algorithms and NLP to automatically create a corpus for enthymeme detection and enthymeme reconstruction. Furthermore, they provide initial baselines to both of these tasks. This way, the main contributions of this paper are: the proposed methodology, the corpus of natural language enthymemes, and the reported baselines in these tasks.

**Reasons To Accept:**

The paper addresses an interesting and original task such as enthymeme identification and reconstruction. Further development in these area can be helpful in the automatic analysis of argumentation since enthymemes are frequently used in argumentative dialogues and speeches. Furthermore, with the proposed method for the automatic generation of an enthymeme corpus, it will be possible to ease the creation of additional corpora for addressing these tasks.

**Reasons To Reject:**

The paper is quite dense and some aspects are not thoroughly described in the current version of the work. I am missing some additional discussion and analysis in sections 3.2 and 3.3. Furthermore, when relying exclusively on probabilistic models to generate a corpus some risks are assumed as pointed out in lines 227-228. A human evaluation of the whole automatic method is missing, and it would be the only way to determine the real quality of the automatically generated corpus instead of independently evaluating the probabilistic models used to previous work approaches.

**Reproducibility:**

4: Could mostly reproduce the results, but there may be some variation because of sample variance or minor variations in their interpretation of the protocol or method.

**Reviewer Confidence:**

4: Quite sure. I tried to check the important points carefully. It's unlikely, though conceivable, that I missed something that should affect my ratings.

---

> ### Author Rebuttal · Authors · 2023-08-28
>
> We thank you for pointing out the importance and originality of both the task and our method.
>
> **Reason to reject #1:** We acknowledge that the approach sections are quite dense. We will use the additional space in the camera-ready version to give more details on candidate enthymeme selection (Section 3.2) and ranking (Section 3.3) in more detail. As part of this, we plan to add a table exemplifying what candidates are filtered out (or not), as well as figure visualizing the PageRank idea.
>
> **Reason to reject #2:** Regarding the overall evaluation, we point out that the manual evaluation in Sections 5.1 and 5.2 of our approach and its components measures the quality of our corpus in terms of creating inadequate enthymemes: By asking the annotators to compare the modified arguments to the original ones, we find that they are of less sufficiency but near-identical naturalness. We will include excerpts of the guidelines in Appendices B and C to clarify this point.

---

### Official Review · Reviewer_432F · 2023-08-04

**Soundness:** 4

**Excitement:**

4: Strong: This paper deepens the understanding of some phenomenon or lowers the barriers to an existing research direction.

**Paper Topic And Main Contributions:**

This paper deals with detection and reconstruction of enthymemes (missing argumentative discourse units), but central is the
construction of a respective corpus that permits the previously mentioned tasks.

The authors propose a regression model for essay quality filtering, over features such as ADU n-grams, sentiment flow, token as POS n-grams, distribution of NEs and metadiscourse markers. Via ablation studies the authors determined the best set of features.
They also take argument length into account.
Naturalness is ranked by BERTs next sentence prediction.

For ranking candidate enthymemes, they use a PageRank-based apporach.

According to the rank, enthymeme candidates are removed from the data in order to produce training data for detection and reconstruction.

This data is then used for enthymeme detection, which is treated as a binary classification task.
The authors fine-tune deBERTa for the classification task.

Enthymeme reconstruction is treated as a mask-infilling task. This time the authors fine-tune BART for the task.

 Main contributions: The authors claim to have produced one of the first (the first?) corpus for systematically exploring enthymeme detection and reconstruction.

**Reasons To Accept:**

The paper is well written and easily understandable. Given the authors' claim is right, this corpus may be the first of his kind and therefore be a legitimate contribution. I am not enough acquainted with this specific topic for any deeper evaluation.

**Reasons To Reject:**

The only thing I could mention is a quote from the authors themselves in the limitation section, saying that the work cannot be compared well since there is no other data available that would permit a comparison.

**Reproducibility:**

4: Could mostly reproduce the results, but there may be some variation because of sample variance or minor variations in their interpretation of the protocol or method.

**Reviewer Confidence:**

2: Willing to defend my evaluation, but it is fairly likely that I missed some details, didn't understand some central points, or can't be sure about the novelty of the work.

---

> ### Author Rebuttal · Authors · 2023-08-28
>
> Thank you for acknowledging the contribution and practical importance of our work. To the best of our knowledge, we indeed introduce the first enthymeme corpus for learner arguments, which is why a comparison to other learner enthymeme datasets is not possible.

---

### Official Review · Reviewer_7rF4 · 2023-08-05

**Typos Grammar Style And Presentation Improvements:** I think overall, the paper is very we…
**Soundness:** 3

**Excitement:**

4: Strong: This paper deepens the understanding of some phenomenon or lowers the barriers to an existing research direction.

**Missing References:**

-

**Paper Topic And Main Contributions:**

This paper presents an automatically constructed corpus for the task of Ethymeme (i.e. implicit argumentative discourse unit, ADU) detection and reconstruction. The paper proposes a pipeline of automated steps to automatically remove central ADUs from argumentative texts ranked high in quality. The pipeline is validated by manual annotation. The paper conducts baseline experiments on the constructed corpus to investigate the feasibility of automatically detecting and reconstructing the removed sentences.

**Questions For The Authors:**

Question A: It being 2023, I'm afraid I have to ask: have you tried any of the publicly available LLMs on your task?

Question B: What would be the (per-document) baseline/trained model accuracy if you evaluate the binary classification task at each possible position (i.e. between two sentences) in a document? In other words, how does the positive/negative performance translate to per-document ethymeme identification accuracy?

Question 3: Why are the agreement scores so low?

**Reasons To Accept:**

The idea is creative, the task seems novel and the automated construction of the dataset is well motivated, documented and evaluated. Overall, the paper is well written and organised. The task and dataset in itself seems well-rounded.

**Reasons To Reject:**

While intrinsically, the paper is coherent, I am not convinced that the proposed proxy-task really represents what the paper claims it represents.

Beyond the human quality judgements, how do we know that applying models trained on these data on learner essays "in the wild" will yield the desired improvements? I understand that a full-scale evaluation of the tool as a writing assistant is out of scope of this paper, but I expected to see at least a sampled manual analysis of the trained model on some real ethymeme examples. What would the classification model detect? What would the reconstruction model generate?

Furthermore, I am not entirely sure why the classification task was formulated as it was. Isn't detecting the position of the inappropriate ethymeme part of the challenge? Instead the task is simplified by providing "True" and "Fake" ethymeme positions and the model essentially only learns to predict whether a sentence is missing or not.

Finally, the agreement scores of the annotators seem rather low, indicating only a slight correlation of the annotators choices.

**Reproducibility:**

4: Could mostly reproduce the results, but there may be some variation because of sample variance or minor variations in their interpretation of the protocol or method.

**Reviewer Confidence:**

4: Quite sure. I tried to check the important points carefully. It's unlikely, though conceivable, that I missed something that should affect my ratings.

---

> ### Author Rebuttal · Authors · 2023-08-28
>
> Thank you for your positive judgment of the novelty, motivation, and thoroughness of our work.
>
> **Reason to reject #1:** We agree that a manual analysis of applying our model on real arguments can provide further insights and will gladly add it, when we revise the paper. As a first illustration what our models detect and reconstruct, we applied them to the following example argument about about the use of credit cards by students:
>
> - `no enthymeme position`
> - One argument that the students use the credit card unwisely.
> - `enthymeme position, reconstruction:` *According to \<R\> the students need to pay off the debts in a short period of time.*
> - They need to do part-time-jobs to earn money to settle the debts.
> - `no enthymeme position`
> - The students invest much of time to do part-time jobs therefore waste a lot of time on their studies.
> - `no enthymeme position`
> - The part-time jobs which is lower payment to the students.
> - `no enthymeme position`
> - The student should manage time effectively to develop a balanced and health life style .
> - `enthymeme position, reconstruction:` *In addition, the students need to learn how to manage their financial affairs in a responsible way.*
>
> Our models identified two enthymeme positions and generated corresponding texts, that makes the connection between credit card usage and depts and the implicit claim of the argument towards the topic explicit.
>
> **Reason to reject #2 + Question B:** The classification task exactly serves to deal with the challenge you raise: Through the formulation we uses, we obtained a balanced dataset with diverse negative instances for training and evaluation. In practice, however, the classifier should be applied to *all* positions. This can lead to multiple enthymeme positions, including the most probable one, which we use for our evaluation. To ensure that this does not negatively effect our results, we will add the performance for including all negative instances to the final version of the paper. Thank you for the suggestion. We expect that the randomness in our training data sufficiently approximates this setting.
>
> **Reason to reject #3 + Question 3:** The agreement scores of the annotators are indeed rather low, as often in crowdsourcing, but similar to related work such as Gurcke et al. (2021). That is why we apply MACE (Hovy et al., 2013) to model each annotator’s confidence in order to obtain more reliable final labels.
>
> **Question A:** As indicated in the paper, the goal of our experiments was to provide baselines rather than to present the find the approach for the task. Hence, we refrained from testing the latest LLMs, but placed value on comparability with previous work that used BART (Gurcke et al., 2021).

---

### Meta-Review · Area_Chair_gx2v · 2023-09-19

**Recommendation:** 3

**Metareview:**

The reviewers agreed on the importance of this novel resources, automatically constructed, for the enthymeme detection and reconstruction tasks. However, they also highlighted some drawbacks to be addressed (i.e., simplification of the task, lack of deeper discussion on the selection and ranking of the enthymemes). The reviewers appreciated the author rebuttal where some issues have been clarified.

---

### Decision · Program_Chairs · 2023-10-07

**Decision:**

Accept-Findings

**Comment:**

The reviewers agreed on the importance of this novel resources, automatically constructed, for the enthymeme detection and reconstruction tasks. However, they also highlighted some drawbacks to be addressed (i.e., simplification of the task, lack of deeper discussion on the selection and ranking of the enthymemes). The reviewers appreciated the author rebuttal where some issues have been clarified.